# Association between METS-IR and Prediabetes or Type 2 Diabetes Mellitus among Elderly Subjects in China: A Large-Scale Population-Based Study

**DOI:** 10.3390/ijerph20021053

**Published:** 2023-01-06

**Authors:** Hui Cheng, Xiao Yu, Yu-Ting Li, Zhihui Jia, Jia-Ji Wang, Yao-Jie Xie, Jose Hernandez, Harry H. X. Wang, Hua-Feng Wu

**Affiliations:** 1School of Public Health, Sun Yat-sen University, Guangzhou 510080, China; 2State Key Laboratory of Ophthalmology, Zhongshan Ophthalmic Center, Sun Yat-sen University, Guangzhou 510060, China; 3Centre for General Practice, The Seventh Affiliated Hospital, Southern Medical University, Foshan 528244, China; 4School of Public Health, Guangzhou Medical University, Guangzhou 510182, China; 5School of Nursing, The Hong Kong Polytechnic University, Hung Hom, Kowloon 852, Hong Kong, China; 6Medicine and Health, EDU Institute of Higher Education, 1320 Kalkara, Malta; 7Green Templeton College, University of Oxford, Oxford OX2 6HG, UK; 8Shishan Community Health Centre of Nanhai, Foshan 528234, China

**Keywords:** METS-IR, prediabetes, T2DM, insulin resistance, cross-sectional study

## Abstract

The metabolic score for insulin resistance (METS-IR) was recently proposed as a non-insulin-based, novel index for assessing insulin resistance (IR) in the Western population. However, evidence for the link between METS-IR and prediabetes or type 2 diabetes mellitus (T2DM) among the elderly Chinese population was still limited. We aimed to investigate the associations between METS-IR and prediabetes or T2DM based on large-scale, cross-sectional, routine physical examination data. In a total of 18,112 primary care service users, an increased METS-IR was independently associated with a higher prevalence of prediabetes (adjusted odds ratio [aOR] = 1.457, 95% confidence interval [CI]: 1.343 to 1.581, *p* < 0.001) and T2DM (aOR = 1.804, 95%CI: 1.720 to 1.891, *p* < 0.001), respectively. The aOR for prediabetes in subjects with the highest quartile of METS-IR was 3.060-fold higher than that in those with the lowest quartile of METS-IR. The aOR for T2DM in subjects with the highest quartile of METS-IR was 6.226-fold higher than that in those with the lowest quartile of METS-IR. Consistent results were obtained in subgroup analyses. Our results suggested that METS-IR was significantly associated with both prediabetes and T2DM. The monitoring of METS-IR may add value to early identification of individuals at risk for glucose metabolism disorders in primary care.

## 1. Introduction

Type 2 diabetes mellitus (T2DM) is a serious, growing, and costly problem for the health system worldwide due to its high prevalence and associated disability and mortality [1]. With the development of the economy, urbanization, aging, and rapid lifestyle transformations in China, the number of diabetic patients among the Chinese population ranks first in the world. The number of T2DM patients in China reached 116 million in 2019, and it is estimated that the number will increase to 141 million by 2045 according to the current trend [2]. Poor blood glucose control in patients with T2DM is common and can lead to multi-system damage, resulting in chronic progressive pathological changes and functional declines in various tissues and organs, such as eyes, kidneys, nerves, hearts, and blood vessels [3,4]. Prediabetes is characterized by slightly elevated blood glucose concentration between normal levels and the diabetes threshold, and there is a high risk of developing T2DM during a relatively short period [5]. Furthermore, the association between prediabetes and risks of cardiovascular and renal diseases have been recognized [6,7,8,9]. Evidence has emphasized that prediabetes should be diagnosed as early as possible to prevent its transition to T2DM and its complications through timely lifestyle adjustment and appropriate pharmacological interventions [7,10].

Insulin resistance (IR) is a key feature of prediabetes and a precursor to the progression of T2DM [11]. Therefore, early and accurate identification of IR has important clinical significance for implementing prevention strategies and optimizing disease management. The euglycemic-hyperinsulinemic clamp (EHC) is the gold standard for direct determination of IR. However, the EHC is invasive, expensive, and complex, and is less frequently used in routine practice [12]. Other fasting insulin-based indexes, such as the homeostatic model assessment for IR (HOMA-IR) and the quantitative insulin sensitivity check index (QUICKI) [13], are also less commonly performed in most primary care facilities. From a public health perspective, non-insulin-based indexes using routine physical examination indicators have been proposed as alternatives for insulin measurements [12,14].

In recent years, the metabolic score for insulin resistance (METS-IR), a novel index considering fasting plasma glucose (FPG), lipid profiles, and obesity index have been used as a simple and high-accuracy index to evaluate IR and predict the occurrence of T2DM [12]. However, there are only a few studies on the associations between METS-IR and blood glucose levels. Thus, our study, based on data from routine examinations in primary care, aims to explore the associations of METS-IR with prediabetes and T2DM among elderly subjects in China.

## 2. Materials and Methods

### 2.1. Study Design and Data Source

We conducted a cross-sectional study among primary care subjects in an urbanised township consisting of 47 communities in Guangdong province, southern China. Target subjects were primary care patients aged 65 years and above who attended free-of-charge, annual check-ups in the context of the community-wide provision of the national basic public health (BPH) service package, including routine disease management and health maintenance. Computerized data were retrieved from the information platform where the health records were documented electronically by public health practitioners at local community health centres (CHCs).

### 2.2. Setting and Participants

The annual check-up was performed onsite in a standardised manner at local CHCs. Demographic information, routine lifestyle behaviours, and disease-related information regarding patients were collected face-to-face by trained public health practitioners. The physical examination and blood taking were performed by clinical staff following standardized procedures. A total of 26,115 primary care service users aged 65 years and older attended the 2018 annual check-up. We excluded individuals who had incomplete health records. This yielded a total of 18,812 subjects who were finally included in the subsequent analyses.

### 2.3. Study Variables and Measurements

A data management checklist of study variables needed for data analysis was jointly reviewed by a research panel consisting of two public health specialists (H.H.X.W. and Y.T.L.), two family medicine professionals (H.F.W. and J.J.W.), and one research epidemiologist (J.H.). Based on patient self-reports, cigarette smoking was classified as never smoker, ex-smoker, or current smoker. The frequency of alcohol drinking was classified as never, sometimes, or often. Subjects who regularly participated in any type of aerobic exercise more than once per week were considered to have exercise habits.

FPG was determined by enzymatic methods. High-density lipoprotein cholesterol (HDL-C) and triglycerides (TG) were directly measured using an automated analyser according to standard operating procedures. All clinical measurements and laboratory tests had internal quality control. Anthropometric parameters were measured using a portable stadiometer and a calibrated digital scale, with participants in light clothing. Blood pressure (BP) was measured in a seated position by routinely validated automatic sphygmomanometers. The arm with the higher pressure was used for BP measurement. The average of two BP readings, 1–2 min apart, was recorded. A venous blood sample at fasting was collected on-site. Being overweight was defined as a body mass index (BMI) of 24 kg/m^2^ or higher [15]. Visceral fat obesity was defined as waist circumference (WC) ≥ 90 cm in men or ≥ 80 cm in women [15]. METS-IR was calculated using the formula (Ln(2×FPG) + TG) × BMI / Ln(HDL-C) [12]. The presence of T2DM was determined if subjects had physician-diagnosed T2DM or had FPG ≥ 126 mg/dL (7.0 mmol/L), whilst prediabetes was defined as FPG of 110–125 mg/dL (6.1–6.9 mmol/L) [16].

### 2.4. Statistical Analysis

Data were presented as *n* (%) or mean ± standard deviation (SD) for categorical and continuous variables, respectively. Subjects were divided into quartiles of METS-IR. We examined the characteristics of subjects stratified by METS-IR quartiles. The one-way analysis of variance (ANOVA) or the chi-square test, where appropriate, was used for between-group comparisons in subjects with different quartiles of METS-IR. Logistic regression analyses were performed to estimate the association between METS-IR and prediabetes or T2DM. The odds ratio (OR) with a 95% confidence interval (CI) was estimated for per-unit increase in METS-IR and for each METS-IR quartile. Three models were constructed: model 1 adjusted for sex, age, and education level; model 2 adjusted for sex, age, education level, cigarette smoking, alcohol drinking, and exercise habits; and model 3 adjusted for sex, age, education level, cigarette smoking, alcohol drinking, exercise habits, SBP, and WC levels. The median value of METS-IR in each quartile was used in the test for linear trend.

Data were also modelled as restricted cubic splines (RCS) with 4 knots, located at the 5th, 35th, 65th, and 95th percentiles, according to Akaike’s information criterion [17], of METS-IR to assess the shape of the association between METS-IR and prediabetes or T2DM. We performed stratified logistic regression analyses in patient subgroups according to age, sex, BMI, WC, and exercise habit to examine the potential modifications and interactions. All models were adjusted for the same set of confounders included in the full-variable model (model 3). All statistical analyses were conducted using SAS (version 9.4, SAS Institute Inc, Cary, NC, USA) and R (version 4.0.2, Core Team, Vienna, Austria). A *p* value of less than 0.05 was considered statistically significant.

### 2.5. Ethics Consideration

Data anonymisation was achieved by removing all patient identifiers from the dataset prior to data analysis. Ethics approval was initially granted and subsequently renewed by the School of Public Health Biomedical Research Ethics Review Committee at Sun Yat-Sen University (Refs: SPH2016027 and SPH2019032) in accordance with the Declaration of Helsinki 2013.

## 3. Results

### 3.1. Characteristics of Study Participants

The mean age of study participants was 73.13 ± 6.19 years, and 7753 (41.21%) were men. The mean METS-IR in all subjects was 35.09 ± 6.94. The detailed characteristics of study participants according to the METS-IR quartiles are presented in Table 1. Patient subgroups with a higher METS-IR index had significantly higher levels of BMI, WC, FPG, SBP, and DBP (*p* < 0.001). The levels of METS-IR also differed significantly across subjects by age, education level, smoking, and exercise habits (*p* < 0.05). There was no significant difference in the distribution of sex and alcohol consumption among subjects in different METS-IR quartiles (*p* > 0.05).

### 3.2. Associations of METS-IR with Prediabetes and T2DM

After adjusting for confounders, METS-IR was significantly associated with prediabetes or T2DM in all three models. In the full-variable model (model 3), a unit increase in baseline METS-IR was shown to be significantly associated with both prediabetes and T2DM (adjusted OR [aOR] = 1.457, 95% CI 1.343–1.581, *p* < 0.001 for prediabetes; aOR = 1.804, 95% CI 1.720–1.891, *p* < 0.001 for T2DM, respectively) (Table 2). When METS-IR was treated as a continuous variable, we observed a non-linear relationship between METS-IR levels and prediabetes or T2DM (Figure 1).

The aOR for prediabetes in subjects with the highest quartile of METS-IR was 3.060-fold higher than that in those with the lowest quartile of METS-IR. In the same model, the aOR for T2DM in subjects with the highest quartile of METS-IR was 6.226-fold higher than that in those with the lowest quartile of METS-IR. When METS-IR was treated as a continuous variable, the statistical significance for the trend of METS-IR was consistent across all different models (*p* for trend < 0.001).

### 3.3. Subgroup Analyses

We found that sex and BMI significantly modified the association between METS-IR and prediabetes (*p* for interaction < 0.05), while age, WC, cigarette smoking, alcohol drinking, and exercise habit did not modify this association (*p* for interaction > 0.05). Meanwhile, we found that BMI significantly modified the association between METS-IR and T2DM (*p* for interaction < 0.05). In addition, age, sex, WC, cigarette smoking, alcohol drinking, and exercise habit did not significantly modify the association between METS-IR and T2DM (*p* for interaction > 0.05) (Figure 2).

## 4. Discussion

### 4.1. Main Findings

In this population-based, cross-sectional study, we explored the association between METS-IR and prediabetes or T2DM, which provided information on early identification for prediabetes and diabetes using METS-IR in primary care settings. After adjusting for potential confounders, our results showed that METS-IR, either as a continuous or a categorical variable, was significantly associated with prediabetes and T2DM among the Chinese elderly population. The subgroup analyses yielded similar results. Our findings showed the ability of METS-IR as a potentially useful indicator for the early identification of individuals with glucose metabolism disorders, which may have implications for primary prevention strategies.

### 4.2. Relationship with Other Studies

The METS-IR has attracted increasing attention in recent years as a novel non-insulin-based index considering FPG, blood lipid profiles, and obesity index. Literature suggests that METS-IR showed good agreement with EHC and frequently sampled intravenous glucose tolerance, and that METS-IR can be used as a simple and accurate index to evaluate IR and predict the occurrence of T2DM in the Western population [12]. A prospective cohort study of Mexican adults showed that individuals with T2DM had higher METS-IR scores at baseline, and that the risk of developing T2DM among people in the highest quartile of METS-IR (i.e., the quarter of the dataset with the highest METS-IR scores) was approximately four times that of people in the lowest METS-IR quartile [12]. A similar conclusion was reached in another prospective cohort study in which data from Japanese non-obese adults demonstrated that the risk of developing T2DM in people with higher METS-IR was nearly 1.2 times that of their counterparts [18]. In our study, we found that an elderly subject was approximately 1.5 times as likely to have prediabetes for a unit increase in METS-IR, and was 1.8 times as likely to have diabetes for a unit increase in METS-IR. Associations of METS-IR with prediabetes and diabetes remained significant in subgroups stratified based on demographic characteristics (age and sex), lifestyles (exercise habit, alcohol drinking, and smoking status), and proxy of abdominal fat mass. This indicated that METS-IR could also apply to a different mix of subjects.

It has been well documented that individuals with prediabetes are also at an increased risk of cardiovascular disease (CVD) and all-cause mortality [6,7,8,9]. According to the American Diabetes Association (ADA), up to 70% of people with prediabetes will eventually develop diabetes [19]. Several trials have shown that the risk of diabetes could be reduced by lifestyle changes and pharmacologic management among individuals with prediabetes [19]. However, there is still a lack of studies on the association between METS-IR and prediabetes. Our results show that METS-IR is significantly correlated with prediabetes overall and in subgroup analysis. This implies the potential of METS-IR as an accessible and reliable indicator for early glucose metabolism disorders in resource-limited settings.

The underlying mechanism of METS-IR associated with prediabetes and T2DM is complicated. Islet β-cell dysfunction and IR are the core pathological features of T2DM [20]. Pancreatic β-cells show weak antioxidant enzyme defence abilities because of the low expression of antioxidant enzymes in islet cells [21]. Evidence suggests that elevated blood glucose can induce islet β-cells to produce reactive oxygen species, leading to oxidative stress and β-cells dysfunction, which, in turn, causes IR and T2DM [21,22]. Some indirect studies also revealed that appropriate antioxidant supplementation can regulate lipid metabolism and improve insulin sensitivity [23,24]. Previous studies have indicated that METS-IR is significantly associated with visceral, intrahepatic, and pancreatic fat content [12], which are known pathophysiological components of IR [25]. Other evidence suggests that long-term high free fatty acid concentration is associated with prolonged exposure to islet triglycerides, which may damage the function of pancreatic β-cells [26,27] and result in impaired glucose-induced insulin secretion [28]. Conversely, IR in the liver can significantly alter the homeostasis of blood glucose and lipids, which can be transformed into hyperglycaemia, impaired lipemia, and increased weight [29].

Our results show that METS-IR is more strongly associated with prediabetes and T2DM in the lower BMI subgroup after adjusting for potential confounders. There could be several explanations. Overweight subjects, who are at higher risk for CVD [30], tend to be screened more regularly with more intensified management of diabetes, which could result in better health outcomes. Another possible explanation is that being overweight might exhibit a significant metabolic reserve, thereby protecting against poor prognoses or health outcomes. People who develop T2DM due to the metabolic stress of being overweight or obese may, thus, have a better prognosis or a lower risk of complications and comorbidities than those with lower BMI due to genetic susceptibility [31,32,33].

With the progressive aging of the population, the concomitant increase in the global prevalence of chronic metabolic disorders has become a public health concern [34,35]. In our present study conducted in the elderly population, the METS-IR score was higher in the younger subgroup (72.17 years for the highest quartile of METS-IR vs. 74.03 years for the lowest quartile of METS-IR). A secondary data analysis of the US National Health and Nutrition Examination Survey (NHANES) also reported a similar pattern [36]. Given the increasing trend of young-onset T2DM [37], further longitudinal investigations could be conducted to explore whether METS-IR levels in early adulthood may be associated with risks for prediabetes and diabetes in later life within the same individual.

### 4.3. Implications for Research and Practice

T2DM remains a major health problem across the globe, which has posed significant challenges to current healthcare and economic systems [38]. Prediabetes is a state characterized by impaired fasting glucose or impaired glucose tolerance, which is also a toxic cardiometabolic state associated with an increased risk for microvascular and macrovascular complications [6,7,8,9,39]. Individuals with prediabetes have a 3- to 10-fold increased risk for T2DM. The identification and treatment of prediabetes are imperative to prevent or delay the progression to T2DM. It has been shown that IR plays an important role in the pathophysiological mechanism for developing glucose metabolism disorders in prediabetes [39]. The American Association of Clinical Endocrinologists (AACE) and the American College of Endocrinology (ACE) advocate that early intervention in the IR stage can effectively lower the risk of CVDs, slow down the progression of the disease, reduce disease burden, improve population indicators, and reduce overall healthcare costs [40]. Therefore, it is of great practical significance to identify a prediction index for the assessment of IR that is precise, easy to measure, and suitable for a different mix of populations, particularly in the delivery of follow-up care in remote or rural settings [41]. In low- and middle-income countries (LMICs) where measurement of insulin level is largely absent in routine check-ups, METS-IR is emerging as the preferred surrogate measure of insulin sensitivity in resource-limited primary care settings.

### 4.4. Strengths and Weaknesses of the Study

To our knowledge, this was the first large-scale, population-based study to explore the associations of METS-IR, a novel and non-insulin-based index for estimation of IR, with both prediabetes and T2DM while evaluating potential non-linear associations. The statistical analyses were systematically performed using METS-IR as a continuous variable and in quartiles following the same methodology to deal with confounding. Consistent results were obtained in subgroup analyses, which enhanced the robustness of our main findings. The study had several limitations. First, we were not able to determine the causality due to the cross-sectional design. Second, we did not directly assess the association between METS-IR and insulin sensitivity, as insulin levels were not routinely assessed in primary care check-ups, especially in LMICs. Third, our study was conducted in older adults, and thus the findings may not be directly generalizable to the younger population. Last but not least, although several confounding factors have been dealt with in the regression models, we were not able to take into account factors, such as family history of metabolic disorders, dietary intake, environmental exposures, and body constitutions [42], that were not adequately captured in the original health check-up records. Given that primary care-based diabetes education may also have an impact on the individual’s health outcome [43], future efforts should be warranted to examine the extent to which the strength of associations between METS-IR and prediabetes or T2DM may vary by levels of engagement in health education.

## 5. Conclusions

The associations of METS-IR with prediabetes and T2DM are explored in this population-based, cross-sectional study. We found that METS-IR was significantly associated with prediabetes and T2DM after adjusting for potential confounders in the Chinese population aged 65 years and above. Consistent results have been obtained in subgroup analyses. Our findings demonstrate the ability of METS-IR as a potentially useful indicator in primary care settings for the early identification of individuals with glucose metabolism disorders, which may have implications for primary prevention strategies. As the prevalence of elevated blood glucose continues to increase, further longitudinal investigations are needed to explore whether METS-IR levels in early adulthood may be associated with risks for prediabetes and diabetes in later life within the same individual.

## Figures and Tables

**Figure 1 ijerph-20-01053-f001:**
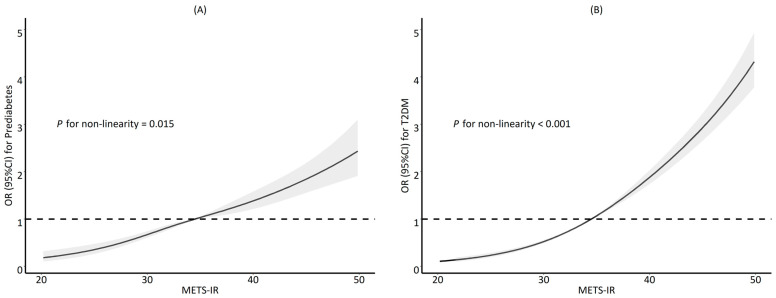
Restricted cubic splines of METS-IR with prediabetes (**A**) and T2DM (**B**). The solid black line represents the fitted curve, and the grey bands represent the 95% confidence interval bands. Model adjusted for age, sex, education level, cigarette smoking, alcohol drinking, exercise habits, systolic blood pressure, and waist circumference. METS-IR, metabolic score for insulin resistance; T2DM, type 2 diabetes mellitus.

**Figure 2 ijerph-20-01053-f002:**
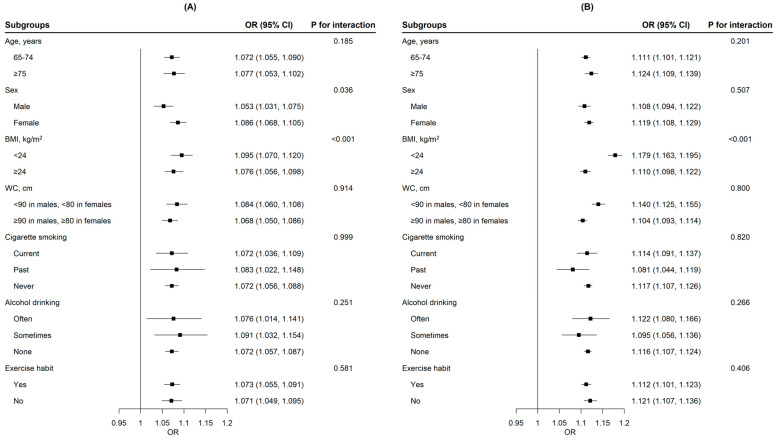
Subgroup analyses of associations of METS-IR with prediabetes (**A**) and T2DM (**B**). Model adjusted for age, sex, education level, annual cigarette smoking, alcohol drinking, exercise habit, systolic blood pressure, and waist circumference. METS-IR, metabolic score for insulin resistance; T2DM, type 2 diabetes mellitus; BMI, body mass index; WC, waist circumference.

**Table 1 ijerph-20-01053-t001:** Characteristics of subjects according to METS-IR quartiles.

Characteristics	Total (*n* = 18812)	Q1 (*n* = 4706)	Q2 (*n* = 4708)	Q3 (*n* = 4694)	Q4 (*n* = 4704)	*p* Value
Age, years	73.13 ± 6.19	74.03 ± 6.90	73.50 ± 6.38	72.81 ± 5.86	72.17 ± 5.34	<0.001
Male, *n* (%)	7753 (41.21)	1883 (40.01)	1973 (41.91)	1919 (40.88)	1978 (42.05)	0.151
Education level, *n* (%)						0.002
Elementary or below	15432 (82.03)	3926 (83.43)	3874 (82.29)	3842 (81.85)	3790 (80.57)	
Middle school	3277 (17.42)	763 (16.21)	809 (17.18)	815 (17.18)	890 (18.92)	
High school or above	103 (0.55)	17 (0.36)	25 (0.53)	37 (0.79)	24 (0.51)	
Cigarette smoking, *n* (%)						<0.001
Current	3167 (16.83)	918 (19.51)	790 (16.78)	729 (15.53)	730 (15.52)	
Past	881 (4.68)	186 (3.95)	239 (5.08)	200 (4.26)	256 (5.44)	
Never	14764 (78.48)	3602 (76.54)	3679 (78.14)	3765 (80.21)	3718 (79.04)	
Alcohol drinking, *n* (%)						
Often	937 (4.98)	267 (5.67)	233 (4.95)	220 (4.69)	217 (4.61)	0.090
Sometimes	968 (5.15)	245 (5.21)	265 (5.63)	230 (4.90)	228 (4.85)	
None	16907 (89.87)	4194 (89.12)	4210 (89.42)	4244 (90.41)	4259 (90.54)	
Exercise habits, *n* (%)						<0.001
Yes	11708 (62.24)	2732 (58.05)	2927 (62.17)	3076 (65.53)	2973 (63.20)	
No	7104 (37.76)	1974 (41.95)	1781 (37.83)	1618 (34.47)	1731 (36.80)	
BMI, kg/m^2^	23.40 ± 3.34	19.80 ± 1.81	22.43 ± 1.52	24.28 ± 1.75	27.09 ± 2.77	<0.001
WC, cm	84.36 ± 9.55	75.91 ± 7.17	81.88 ± 6.86	86.64 ± 6.97	93.02 ± 7.73	<0.001
FPG, mg/dL	99.71 ± 33.01	90.86 ± 19.27	95.10 ± 24.84	100.56 ± 30.05	112.35 ± 46.99	<0.001
SBP, mmHg	129.61 ± 14.08	126.82 ± 13.78	129.22 ± 12.89	130.57 ± 15.90	131.84 ± 13.08	<0.001
DBP, mmHg	77.33 ± 9.21	76.13 ± 13.26	76.99 ± 7.37	77.69 ± 7.17	78.48 ± 7.37	<0.001
METS-IR index	35.09 ± 6.94	26.90 ± 2.48	32.39 ± 1.25	36.79 ± 1.37	44.30 ± 4.56	<0.001

Note: METS-IR, metabolic score for insulin resistance; BMI, body mass index; WC, waist circumference; SBP, systolic blood pressure; DBP, diastolic blood pressure; FPG, fasting plasma glucose. To convert glucose values from mg/dL to mmol/L multiply by 0.0555. Quartiles of METS-IR index: Q1, ≤30.15; Q2, 30.16 to 34.49; Q3, 34.50 to 39.30; Q4, ≥ 39.31.

**Table 2 ijerph-20-01053-t002:** The associations of METS-IR with prediabetes and T2DM.

Variables	Crude Model	Model 1	Model 2	Model 3
	OR (95%CI)	aOR (95%CI)	aOR (95%CI)	aOR (95%CI)
Prediabetes				
METS-IR, per unit increase	1.426 (1.341, 1.517) *	1.444 (1.356, 1.536) *	1.446 (1.359, 1.539) *	1.457 (1.343, 1.581) *
METS-IR, quartiles				
Quartile 1 (≤29.56)	Reference	Reference	Reference	Reference
Quartile 2 (29.57 to 33.71)	1.522 (1.211, 1.913) *	1.538 (1.223, 1.933) *	1.541 (1.225, 1.937) *	1.537 (1.215, 1.945) *
Quartile 3 (33.72 to 38.27)	2.044 (1.645, 2.541) *	2.090 (1.680, 2.600) *	2.094 (1.682, 2.606) *	2.082 (1.638, 2.647) *
Quartile 4 (≥38.28)	2.930 (2.381, 3.606) *	3.036 (2.463, 3.743) *	3.061 (2.481, 3.775) *	3.060 (2.353, 3.978) *
*p* for trend	1.072 (1.059, 1.086) *	1.075 (1.062, 1.089) *	1.075 (1.062, 1.089) *	1.075 (1.058, 1.093) *
T2DM				
METS-IR, per unit increase	1.786 (1.722, 1.852) *	1.809 (1.744, 1.877) *	1.804 (1.739, 1.872) *	1.804 (1.720, 1.891) *
METS-IR, quartiles				
Quartile 1 (≤30.15)	Reference	Reference	Reference	Reference
Quartile 2 (30.16 to 34.49)	2.018 (1.761, 2.314) *	2.044 (1.782, 2.343) *	2.025 (1.766, 2.322) *	2.027 (1.761, 2.332) *
Quartile 3 (34.50 to 39.30)	3.137 (2.754, 3.573) *	3.213 (2.820, 3.662) *	3.162 (2.774, 3.605) *	3.168 (2.747, 3.652) *
Quartile 4 (≥39.31)	6.034 (5.326, 6.836) *	6.275 (5.533, 7.117) *	6.204 (5.470, 7.038) *	6.226 (5.336, 7.265) *
*p* for trend	1.117 (1.109, 1.125) *	1.120 (1.112, 1.127) *	1.119 (1.111, 1.127) *	1.119 (1.109, 1.129) *

Note: Model 1 adjusted for sex, age, and education level; Model 2 adjusted for Model 1 + cigarette smoking, alcohol drinking, and exercise habits; Model 3 adjusted for Model 2 + SBP and WC. METS-IR, metabolic score for insulin resistance; T2DM, type 2 diabetes mellitus; BMI, body mass index; SBP, systolic blood pressure; WC, waist circumference. * *p* value < 0.001.

## Data Availability

The data presented in this study are available on reasonable request from the corresponding author.

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
