# Peer review of "Association between METS-IR and Prediabetes or Type 2 Diabetes Mellitus among Elderly Subjects in China: A Large-Scale Population-Based Study"

_ijerph, 2023, doi:10.3390/ijerph20021053_

Round 1
Reviewer 1 Report
This reviewer considers that the author could take advantadge of their patents recruitment facilities. identify the potential impact of education (general ad specific diabetes education) on their final outcomes.
Author Response
Dear Reviewer 1,
We would like to thank you for your most favourable comments. We have revised our paper according to your valuable comments, with revisions shown in track changes in our manuscript. The point-by-point responses are shown as below.
Point 1: This reviewer considers that the author could take advantadge of their patents recruitment facilities. identify the potential impact of education (general ad specific diabetes education) on their final outcomes.
Response 1: Thank you for your favourable comments. We agree with your opinion that primary care-based diabetes education may also have an impact on the individual’s health outcome. As this was not measured in the present study, we have added into the study limitation that future efforts are warranted to examine the extent to which the strength of associations between METS-IR and prediabetes or T2DM may vary by levels of engagement in health education. Please see lines 302-306 for the revision.
We thank you again for all your valuable comments.
Reviewer 2 Report
Describe Q1-Q4 below table 1
Is there any data about METS-IR associated with family history of metabolic disorders? Does your study show it?
Author Response
Dear Reviewer 2,
We would like to thank you for your most favourable comments. We have revised our paper according to your valuable comments, with revisions shown in track changes in our manuscript. The point-by-point responses are shown as below.
Point 1: Describe Q1-Q4 below table 1
Response 1: Thank you for your valuable advice. We have added the description of METS-IR quartiles (Q1-Q4) into the footnote of Table 1 as follows: Quartiles of METS-IR index: Q1, ≤ 30.15; Q2, 30.16 to 34.49; Q3, 34.50 to 39.30; Q4, ≥ 39.31. Please see the footnotes beneath Table 1 for the revision. Thank you.
Point 2: Is there any data about METS-IR associated with family history of metabolic disorders? Does your study show it?
Response 2: We appreciate your valuable suggestions. Unfortunately, the information on family history of metabolic diseases was not captured in the original health records and thus we were not able to take this confounding factor into account in the analysis. We have added this point into the discussion of study limitation. Please see lines 300-301 for the revision.
We thank you again for all your valuable comments.
Reviewer 3 Report
To the authors: you are presenting results about the use of the new index to evaluate insulin resistance in a population without a direct measure of it. I find your results interesting. It could be more valuable the conclusion if you could include younger people If you could focus the discussion a little more on your results using the index value for each group could be easier to understand for readers no so trained in epidemiological studies.
Author Response
Dear Reviewer 3,
We would like to thank you for your most favourable comments. We have revised our paper according to your valuable comments, with revisions shown in track changes in our manuscript. The point-by-point responses are shown as below.
Point 1: To the authors: you are presenting results about the use of the new index to evaluate insulin resistance in a population without a direct measure of it. I find your results interesting.
Response 1: Thank you for your most favourable comments.
Point 2: If you could focus the discussion a little more on your results using the index value for each group could be easier to understand for readers no so trained in epidemiological studies.
Response 2: Thank you for your favourable comments. We agree with your opinion and we have now rephrased the sentences in the Discussion section regarding the association between METS-IR and health outcomes as follows.
A prospective cohort study of Mexican adults showed that individuals with T2DM had higher METS-IR scores at baseline, and that the risk of developing T2DM among people in the highest quartile of METS-IR (i.e., the quarter of the dataset with the highest METS-IR scores) was approximately four times that of people in the lowest METS-IR quartile [12]. A similar conclusion was reached in another prospective cohort study in which data from Japanese non-obese adults demonstrated that the risk of developing T2DM in people with higher METS-IR was nearly 1.2 times that of their counterparts [18]. In our study, we found that an elderly subject was approximately 1.5 times as likely to have prediabetes for a unit increase in METS-IR, and was 1.8 times as likely to have diabetes for a unit increase in METS-IR.
Please see lines 212-222 for the revision. Thank you.
Point 3: It could be more valuable the conclusion if you could include younger people.
Response 3: Thank you for your favourable comment and valuable advice. Our data were retrieved from primary care check-up for individuals aged 65 years and over and thus it is a shame that the younger people were not included in the analysis. However, in our present study conducted in the elderly population, the METS-IR score was higher in the younger subgroup (72.17 years for the highest quartile of METS-IR vs. 74.03 years for the lowest quartile of METS-IR). A secondary data analysis of the US National Health and Nutrition Examination Survey (NHANES) also reported a similar pattern [36]. Given the increasing trend of young-onset T2DM [37], further longitudinal investigations could be conducted to explore whether METS-IR levels in early adulthood may be associated with risks for prediabetes and diabetes in later life within the same individual.
Please see lines 260-267 for the revision.
We thank you again for all your valuable comments.